# Meta-learning from Tasks with Heterogeneous Attribute Spaces

**Tomoharu Iwata**
NTT Communication Science Laboratories
tomoharu.iwata.gy@hco.ntt.co.jp

**Atsutoshi Kumagai**
NTT Software Innovation Center
atsutoshi.kumagai.ht@hco.ntt.co.jp

## Abstract

We propose a heterogeneous meta-learning method that trains a model on tasks with various attribute spaces, such that it can solve unseen tasks whose attribute spaces are different from the training tasks given a few labeled instances. Although many meta-learning methods have been proposed, they assume that all training and target tasks share the same attribute space, and they are inapplicable when attribute sizes are different across tasks. Our model infers latent representations of each attribute and each response from a few labeled instances using an inference network. Then, responses of unlabeled instances are predicted with the inferred representations using a prediction network. The attribute and response representations enable us to make predictions based on the task-specific properties of attributes and responses even when attribute and response sizes are different across tasks. In our experiments with synthetic datasets and 59 datasets in OpenML, we demonstrate that our proposed method can predict the responses given a few labeled instances in new tasks after being trained with tasks with heterogeneous attribute spaces.

## 1 Introduction

Humans can learn from their various experiences and use such knowledge for new tasks that are related to but different from their experiences. In contrast, since machine learning methods are usually trained on a specific task, they can only be used for that task. Therefore, we need to prepare a large amount of training data when we tackle new tasks. However, preparing sufficient training data requires high cost and is time-consuming in real-world applications. For such problems, much attention has been paid to few-shot learning, which is a framework for obtaining models that can learn from fewer examples. Although many few-shot learning methods have recently been proposed [31, 22, 18, 28, 7, 13, 9], the existing methods assume that all training and target tasks share the same attribute space, and cannot handle tasks with heterogeneous attribute spaces. Here, heterogeneous attribute spaces denote that their attribute spaces are different from each other [38, 20, 37]. This limitation prevents us from learning with a wide variety of tasks, which might contain useful knowledge to cope with new tasks. Although tasks with heterogeneous attribute spaces have been considered in domain adaptation, or transfer learning [20, 38, 32, 34, 17, 33, 37], these methods assume only two tasks, and require target datasets for training.

In this paper, we propose a few-shot learning method for tasks with heterogeneous attribute spaces. The proposed method trains a model on tasks with various attribute spaces, such that it can solve unseen tasks whose attribute spaces are different from the training tasks given a few labeled instances. Figure 1 shows our problem formulation. Our model predicts the response of an attribute vector, which is called a query, given a few instances, which are called a support set, where the number of attributes can be arbitrary, and the target task is different from the training tasks. Our model assumes latent attribute vectors for the representation of each attribute, and latent response vectors for the representation of each response. The latent attribute vectors and latent response vectors are inferred

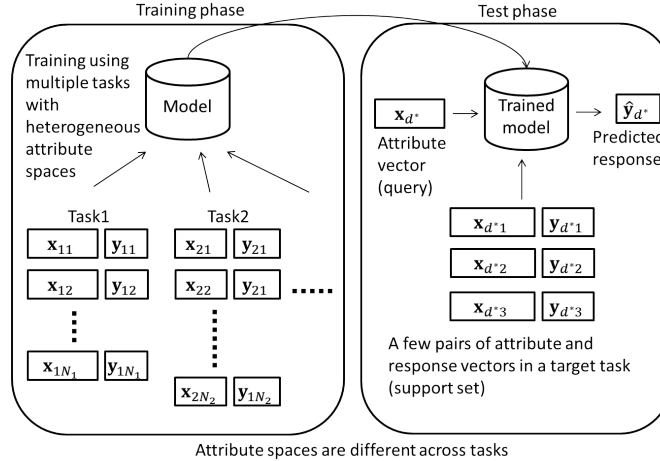

Figure 1: Our problem formulation: In a training phase, our model is trained from various tasks with heterogeneous attribute spaces. In a test phase, the trained model predicts a response of an attribute vector using a few labeled data, where the attribute space of the target task are different from those of the training tasks.

from the support set using an inference network. We design the inference network by effectively combining multiple neural networks that accept variable length inputs, so that the latent vectors contain information about their own empirical marginal distribution and relationships between other attributes and responses. A prediction network predicts responses of queries using the latent vectors, by which we can make predictions considering the properties of the attributes and responses that are specific to each task. The parameters of the neural networks, which are shared across all tasks, are estimated by minimizing the expected test error of the response predictions over tasks with heterogeneous attribute spaces.

The following are the main contributions of this paper: 1) To the best of our knowledge, our work is the first attempt at few-shot learning with tasks with heterogeneous attribute spaces. 2) We propose a neural network-based model for obtaining the representations of attributes, responses, and instances from datasets with any number of attributes, responses, and instances. 3) We empirically demonstrate that the proposed method performs well in few-shot learning on tasks with heterogeneous attribute spaces. The proposed method can be used for situations where we have data from multiple tasks that are related to the target task but their attributes are different across tasks. For example, consider anomaly detection for various machines in various factories. Although the attributes are different across machines since they uses different sesing devices, there are related machines. We can detect anomalies for a new machine in a new factory with only a few labeled data by utilizing data of existing machines.

## 2 Related work

A number of frameworks have been proposed for few-shot learning, or meta-learning [27, 3], including recurrent network-based [22], optimizer-based [1], nearest neighbor-based [31, 28, 2], gradient descent-based [7, 18, 12, 8, 26, 35], and generative model-based methods [5, 9, 11, 10, 4, 23, 25]. These existing few-shot or meta-learning methods cannot learn from tasks with heterogeneous attribute spaces. Our model is related to conditional neural processes [9] in the sense that both types of methods infer task representations using neural networks. The neural process represents a task by a single vector. On the other hand, our model represents a task by a set of latent attribute vectors and latent response vectors to handle tasks with heterogeneous attribute spaces. Tasks with heterogeneous attribute spaces have been considered in domain adaptation, or transfer learning [20, 38, 32, 34, 17, 33, 37], where source and target domains are assumed to have different attribute spaces. However, these domain adaptation methods assume only two tasks (source and target) and require target datasets for training. In contrast, the proposed method can handle more than two tasks and does not use target

datasets for training. Although some domain adaptation methods do not require target datasets for training [15], they are inapplicable to tasks with heterogeneous attribute spaces.

# 3 Proposed method

## 3.1 Problem formulation

Suppose that we are given datasets in multiple tasks with heterogeneous attribute spaces $\{\mathcal{D}_d\}_{d=1}^D$ at a training phase, where $\mathcal{D}_d = \{(\mathbf{x}_{dn}, \mathbf{y}_{dn})\}_{n=1}^{N_d}$ is the set of the pairs of observed attribute and response vectors in task $d$, $\mathbf{x}_{dn} \in \mathbb{R}^{I_d}$ is the observed attribute vector of the $n$th instance, $\mathbf{y}_{dn} \in \mathbb{R}^{J_d}$ is the observed response vector of the $n$th instance, $N_d$ is the number of instances, $I_d$ is the number of attributes, and $J_d$ is the number of responses. The attributes and responses in a task can be different from those in other tasks. The numbers of instances, attributes and responses can be different across tasks $N_d \neq N_{d'}$, $I_d \neq I_{d'}$, and $J_d \neq J_{d'}$.

At a test phase, we are given dataset on a target task $\mathcal{D}_{d^*} = \{(\mathbf{x}_{d^*n}, \mathbf{y}_{d^*n})\}_{n=1}^{N_{d^*}}$, which is called a support set. Here, the number of instances $N_{d^*}$ is small. The target task is not contained in the given training tasks $d^* \notin \{1, \dots, D\}$, and the numbers of attributes and responses of the target task can be different from those of the training tasks. We want to predict response $\mathbf{y}_{d^*}$ for observed attribute vector $\mathbf{x}_{d^*}$, which is called a query, in the target task.

## 3.2 Model

Our model predicts response $\hat{\mathbf{y}}$ of query attribute vector $\mathbf{x}$ using support set $\mathcal{S} = \{(\mathbf{x}_n, \mathbf{y}_n)\}_{n=1}^N$, where $\mathbf{x}_n = (x_{ni})_{i=1}^I$ is the $I$-dimensional observed attribute vector, $\mathbf{y}_n = (y_{nj})_{j=1}^J$ is the $J$-dimensional observed response vector, and we omit task index $d$ for simplicity. We assume that $x_{ni}$ and $y_{nj}$ are scalar numerical values although our framework is applicable to categorical values using one-hot encoding. Figure 2 shows the procedure for response prediction with our model. With our model, first, latent attribute vectors, $\mathbf{V} = \{\mathbf{v}_i\}_{i=1}^I$, and latent response vectors, $\mathbf{C} = \{\mathbf{c}_j\}_{j=1}^J$, are obtained using support set $\mathcal{S}$ by an inference network, as described in Section 3.2.1, where $\mathbf{v}_i$ is the representation of the $i$th attribute, and $\mathbf{c}_j$ is the representation of the $j$th response. Then with a prediction network, latent instance vector $\mathbf{z}$ is obtained using latent attribute vectors $\mathbf{V}$ and query attribute vector $\mathbf{x}$, and response $\hat{\mathbf{y}}$ is predicted using latent response vectors $\mathbf{C}$ and latent instance vector $\mathbf{z}$, as described in Section 3.2.2.

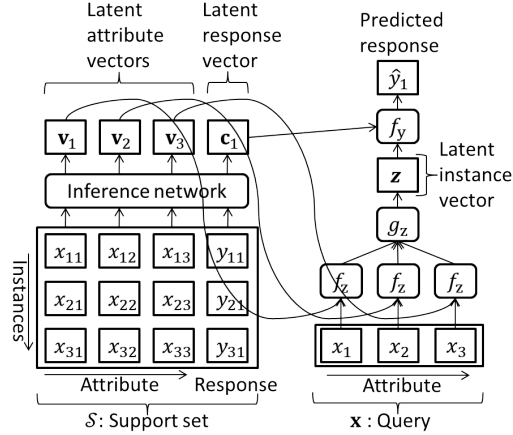

Figure 2: Our model for response prediction of query $\mathbf{x}$ given support set $\mathcal{S}$.

## 3.2.1 Inference network

First, we calculate initial attribute representation $\bar{\mathbf{v}}_i$ and initial response representation $\bar{\mathbf{c}}_j$ using support set $\mathcal{S}$:

$$\bar{\mathbf{v}}_i = g_{\bar{\mathbf{v}}}\left(\frac{1}{N}\sum_{n=1}^N f_{\bar{\mathbf{v}}}(x_{ni})\right), \qquad \bar{\mathbf{c}}_j = g_{\bar{\mathbf{c}}}\left(\frac{1}{N}\sum_{n=1}^N f_{\bar{\mathbf{c}}}(y_{nj})\right), \tag{1}$$

where $f_{\bar{\mathbf{v}}}$, $g_{\bar{\mathbf{v}}}$, $f_{\bar{\mathbf{c}}}$, and $g_{\bar{\mathbf{c}}}$ are feed-forward neural networks. Eq. (1) is a permutation invariant neural network [36] since the summation is invariant even when the elements are permuted, and it can take any number of instances as input. The permutation invariance is appropriate since the representation should not depend on the order of the instances in the support set. By using set of values for the attribute $\{x_{ni}\}_{n=1}^N$ (response $\{y_{nj}\}_{n=1}^N$) in Eq. (1), the information about the empirical

marginal distribution of the attribute can be encoded in initial attribute representation $\bar{\mathbf{v}}_i$ (response representation $\bar{\mathbf{c}}_i$).

The initial attribute and response representations, $\bar{\mathbf{v}}_i$ and $\bar{\mathbf{c}}_j$, do not contain information about the relationship with other attributes and responses, since they are calculated only using their values, $\{x_{ni}\}_{n=1}^N$ or $\{y_{nj}\}_{n=1}^N$. To encode the information about all the attributes and responses of the instance, we calculate the representation for the $n$th instance, $\mathbf{u}_n$, using the initial attribute and response representations:

$$\mathbf{u}_n = g_{\mathrm{u}}\left(\frac{1}{I}\sum_{i=1}^I f_{\mathrm{u}}([\bar{\mathbf{v}}_i, x_{ni}]) + \frac{1}{J}\sum_{j=1}^J f_{\mathrm{u}}([\bar{\mathbf{c}}_j, y_{nj}])\right),\tag{2}$$

where $f_{\mathrm{u}}$ and $g_{\mathrm{u}}$ are feed-forward neural networks, and $[\cdot,\cdot]$ represents the concatenation. With Eq. (2), we obtain a fixed-size vector that represents an instance even when the numbers of attributes and responses are different. By concatenating the representations and their values, $[\bar{\mathbf{v}}_i, x_{ni}]$ and $[\bar{\mathbf{c}}_j, y_{nj}]$, we incorporate information on each attribute and response with a permutation invariant neural network in Eq. (2).

Next, we calculate attribute representation $\mathbf{v}_i$ and response representation $\mathbf{c}_j$ using the instance representations:

$$\mathbf{v}_i = g_{\mathrm{v}}\left(\frac{1}{N}\sum_{n=1}^N f_{\mathrm{v}}([\mathbf{u}_n, x_{ni}])\right), \qquad \mathbf{c}_j = g_{\mathrm{c}}\left(\frac{1}{N}\sum_{n=1}^N f_{\mathrm{c}}([\mathbf{u}_n, y_{nj}])\right),\tag{3}$$

where $f_{\mathrm{v}}$, $g_{\mathrm{v}}$, $f_{\mathrm{c}}$, and $g_{\mathrm{c}}$ are feed-forward neural networks. By concatenating instance representation $\mathbf{u}_n$ and the $i$th attribute's value $x_{ni}$, we encode the relationship of the $i$th attribute to the other attributes and responses, since $\mathbf{u}_n$ contains the information about all attributes and responses of the instance. We can obtain a deep version of attribute and response vectors by iterating Eqs. (2,3), where $\bar{\mathbf{v}}_i$ and $\bar{\mathbf{c}}_j$ in Eq. (2) are replaced by $\mathbf{v}_i$ and $\mathbf{c}_j$ in Eq. (3) at the previous step.

### 3.2.2 Prediction network

Given observed attribute vector $\mathbf{x} = (x_i)_{i=1}^I$ as a query, we obtain latent instance vector $\mathbf{z}$, which is the query's representation, using latent attribute vectors $\mathbf{V}$:

$$\mathbf{z} = g_{\mathrm{z}}\left(\frac{1}{I}\sum_{i=1}^I f_{\mathrm{z}}([\mathbf{v}_i, x_i])\right),\tag{4}$$

where $f_{\mathrm{z}}$ and $g_{\mathrm{z}}$ are feed-forward neural networks. We encode information about the value for each attribute by concatenating latent attribute vector $\mathbf{v}_i$ and its value $x_i$ with a permutation invariant neural network while allowing variable length inputs. Since responses are not given for queries, the latent response vectors are not used.

We predict response $\hat{\mathbf{y}}$ on query $\mathbf{x}$ using latent instance vector $\mathbf{z}$ and latent response vectors $\mathbf{C}$. In particular, the $j$th response variable is predicted by

$$\hat{y}_j(\mathbf{x}, \mathcal{S}; \boldsymbol{\Phi}) = f_{\mathrm{y}}([\mathbf{c}_j, \mathbf{z}]),\tag{5}$$

where $f_{\mathrm{y}}$ is a feed-forward neural network that outputs a scalar value. The prediction depends on support set $\mathcal{S}$ and parameters $\boldsymbol{\Phi}$ of the following neural networks: $f_{\bar{\mathrm{v}}}$, $g_{\bar{\mathrm{v}}}$, $f_{\bar{\mathrm{c}}}$, $g_{\bar{\mathrm{c}}}$, $f_{\mathrm{u}}$, $g_{\mathrm{u}}$, $f_{\mathrm{v}}$, $g_{\mathrm{v}}$, $f_{\mathrm{c}}$, $g_{\mathrm{c}}$, $f_{\mathrm{z}}$, $g_{\mathrm{z}}$, and $f_{\mathrm{y}}$.

### 3.3 Training

We estimate neural network parameters $\boldsymbol{\Phi}$ by minimizing the following loss that is calculated from randomly generated support sets $\mathcal{S}$ and query sets $\mathcal{Q}$ using the given training datasets $\{\mathcal{D}_d\}_{d=1}^D$:

$$\hat{\boldsymbol{\Phi}} = \arg\min_{\boldsymbol{\Phi}} \mathbb{E}_{\mathcal{D}_d}[\mathbb{E}_{(\mathcal{S}, \mathcal{Q})\sim\mathcal{D}_d}[E(\mathcal{Q}|\mathcal{S}; \boldsymbol{\Phi})]],\tag{6}$$

where $\mathbb{E}$ represents an expectation,

$$E(\mathcal{Q}|\mathcal{S}; \boldsymbol{\Phi}) = \frac{1}{N_{\mathrm{Q}} J_{\mathrm{Q}}} \sum_{(\mathbf{x}, \mathbf{y})\in\mathcal{Q}} \sum_{j=1}^{J_{\mathrm{Q}}} \| y_j - \hat{y}_j(\mathbf{x}, \mathcal{S}; \boldsymbol{\Phi}) \|^2,\tag{7}$$

---

**Algorithm 1** Training procedure of our model: $\mathrm{RandomSample}(\mathcal{S}, N)$ generates a set of $N$ elements chosen uniformly at random from set $\mathcal{S}$ without replacement.

---

**Input:** Datasets from tasks with heterogeneous attribute spaces $\{\mathcal{D}_d\}_{d=1}^{D}$, number of support instances $N_S$, number of query instances $N_Q$, batch size $B$
**Output:** Trained model parameters $\boldsymbol{\Phi}$
 1: **while** End condition is satisfied **do**
 2: &emsp; Initialize loss, $J \leftarrow 0$
 3: &emsp; Select task indices for a mini batch, $\mathcal{M} \leftarrow \mathrm{RandomSample}(\{1, \cdots, D\}, B)$
 4: &emsp; **for** $d \in \mathcal{M}$ **do**
 5: &emsp;&emsp; Generate support set, $\mathcal{S} \leftarrow \mathrm{RandomSample}(\mathcal{D}_d, N_S)$
 6: &emsp;&emsp; Generate query set, $\mathcal{Q} \leftarrow \mathrm{RandomSample}(\mathcal{D}_d, N_Q)$
 7: &emsp;&emsp; Calculate loss by Eq. (7), $J \leftarrow J + E(\mathcal{Q}|\mathcal{S}; \boldsymbol{\Phi})$, and its gradients
 8: &emsp; **end for**
 9: &emsp; Update model parameters $\boldsymbol{\Phi}$ using loss $J$ and its gradient
10: **end while**

---

is the loss on the query set given the support set, $N_Q$ is the number of instances in the query set, and $J_Q$ is the number of responses in the query set. The training procedure of our model is shown in Algorithm 1.

### 3.4 Classification

In previous subsections, we assume regression tasks, where responses are numerical values. For classification tasks, where responses are categorical values, we use a nearest neighbor-based approach. In particular, a prototype for each class is obtained by the mean of the latent instance vectors of the support instances that belong to the class: $\hat{\mathbf{z}}_{jk} = \frac{1}{|\mathcal{S}_{jk}|} \sum_{(\mathbf{x},y)\in\mathcal{S}_{jk}} g_z\left(\frac{1}{I}\sum_{i=1}^{I} f_z([\mathbf{v}_i, x_i])\right)$, where $\hat{\mathbf{z}}_{jk}$ is the prototype of class $k$ in the $j$th response, $\mathcal{S}_{jk} = \{\mathbf{x}_n | y_{jn} = k, \mathbf{x}_n \in \mathcal{S}\}$ is the support instances that belongs to class $k$ in the $j$th response, and the latent instance vectors are calculated based on Eq. (4). Then, the class probability is calculated by the distance between the query latent instance vector and prototypes: $\hat{p}(y_j = k|\mathbf{x}, \mathcal{S}; \boldsymbol{\Phi}) = \frac{\exp(-\|\mathbf{z}-\hat{\mathbf{z}}_{jk}\|^2)}{\sum_{k'=1}^{K_j} \exp(-\|\mathbf{z}-\hat{\mathbf{z}}_{jk'}\|^2)}$, where $\mathbf{z}$ is the query's latent instance vector in Eq. (4), and $K_j$ is the number of classes in the $j$th response. We use the cross-entropy loss for classification tasks instead of the mean squared error in Eq. (7).

## 4 Experiments

### 4.1 Synthetic data

**Data** We first evaluated the proposed method on simple synthetic regression tasks with one- or two-dimensional attribute spaces and a one-dimensional response space. One third of the tasks were generated from a one-dimensional linear model, $y = w_d x$, one third were generated from a one-dimensional sine curve, $y = \sin(x + 3w_d)$, and the remaining tasks were generated from the following two-dimensional model, $y = w_{d1} x_1 + \sin(x_2 + 3w_{d2})$. Attributes $x$, $x_1$, and $x_2$ were uniform randomly generated from $[-3, 3]$, and task-specific model parameters $w_d$, $w_{d1}$, and $w_{d2}$ were uniform randomly generated from $[-1, 1]$. We generated 10,000 training, 30 validation, and 300 target tasks. The number of support instances was $N_S = 5$, and the number of query instances was $N_Q = 27$.

**Proposed method settings** We used three-layered feed-forward neural networks with 32 hidden units for all neural networks. The parameters were shared between the following pairs of neural networks: $(f_{\bar{v}}, f_{\bar{c}})$, $(g_{\bar{v}}, g_{\bar{c}})$, $(f_v, f_c)$, $(g_v, g_c)$, and The number of units at the output layer for $f_y$ was one, and it was 32 for the other neural networks. We used rectified linear unit, $\mathrm{ReLU}(x) = \max(0, x)$, for the activation. We optimized using Adam [14] with learning rate $10^{-3}$ and dropout rate 0.1. The validation data were used for early stopping. The batch size was $B = 256$. We implemented the proposed method with PyTorch [21].

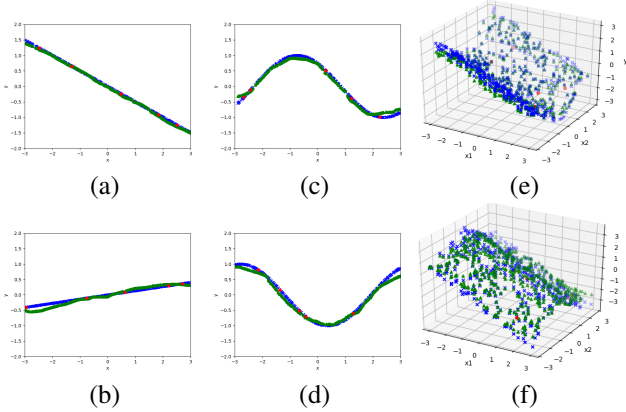

(a)        (c)        (e)

(b)        (d)        (f)

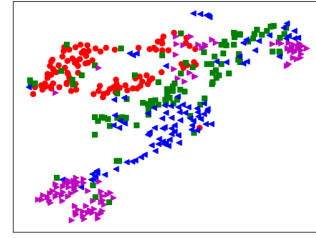

Figure 3: Prediction by the proposed method for target tasks in the synthetic datasets: Red circles are five target support instances, blue crosses are true target query instances, and green triangles are predicted target query instances with the proposed method.

Figure 4: t-SNE visualization of latent attribute vectors $\mathbf{v}_{di}$ for target support sets in the synthetic datasets: The color indicates the attribute index; red: $x$ in $y = w_d x$, green: $x$ in $y = \sin(x + 3w_d)$, blue and magenta: $x_1$ and $x_2$ in $y = w_{d1}x_1 + \sin(x_2 + 3w_{d2})$.

**Results**    The results by the proposed method with the six target tasks are shown in Figure 3. The proposed method appropriately learned two-dimensional linear (a, b) and nonlinear (c, d) relationships as well as a three-dimensional relationship with a single model. Figure 4 shows the visualization of latent attribute vectors $\mathbf{v}_{di}$ inferred from each target support set in the synthetic data, where the latent attribute vectors in all the tasks were simultaneously embedded in the same two-dimensional space by t-SNE [19]. The latent attribute vectors with the same attribute property were closely located to each other.

## 4.2   OpenML datasets

**Data**    We next evaluated the proposed method on tasks with heterogeneous attribute spaces obtained from OpenML [29], which is an open online platform for machine learning that holds various tasks. Using a python API for OpenML [6], we obtained datasets based on the following conditions: the number of instances was between 10 and 300, the number of attributes was between 2 and 30, and all the attributes were numerical values. Then we omitted datasets that had the same number of instances and the same number of attributes, and we obtained 59 datasets in total. The number of instances and attributes for each dataset is shown in the supplemental material. We normalized the values for each attribute with a mean of zero and a variance of one. The last attribute was used as the response for each dataset. We randomly split the 59 tasks into 37 training, 5 validation, and 17 target tasks. The number of support instances was $N_S = 3$, and the number of query instances was $N_Q = 29$.

**Proposed method settings**    We used the same neural network architecture with the synthetic data experiments. The number of epochs was 100,000, and the batch size was $B = 37$. Since an attribute in a task can resemble a response in another task, we randomly selected attributes and a response from a mix of attributes and responses in each task for the support and query sets for each training iteration.

**Comparing methods**    We compared the proposed method with the following 13 methods: deep set [36] (DS), DS with fine-tuning (DS+FT), DS with model-agnostic meta-learning [7] (DS+MAML), conditional neural process [9] (NP), NP with fine-tuning (NP+FT), NP with model-agnostic meta-learning (NP+MAML), linear regression with L2 regularization (Ridge), linear regression with L1 regression (Lasso), Bayesian ridge regression (BR), kernel ridge regression with a linear kernel (KR), Gaussian process regression with an RBF kernel (GP), neural network (NN), and the mean of the support set (Mean). DS, DS+FT, DS+MAML, NP, NP+FT, NP+MAML, and the proposed method used the training datasets for training. Ridge, Lasson, BR, KR, GP, NN, and mean did not use the training datasets, but used the support set of the target task for training. The details of the comparing methods are described in the supplemental material.

Table 1: Averaged mean squared errors and standard errors on the target tasks with the OpenML datasets. Values in bold typeface are not statistically different at 5% level from the best performing method in each row by a paired t-test.

| Method | MSE | Method | MSE | Method | MSE |
|---|---|---|---|---|---|
| Ours | $\mathbf{0.788 \pm 0.011}$ | NP+FT | $0.907 \pm 0.013$ | KR | $0.828 \pm 0.021$ |
| DS | $0.896 \pm 0.011$ | NP+MAML | $0.845 \pm 0.012$ | GP | $1.113 \pm 0.112$ |
| DS+FT | $0.887 \pm 0.011$ | Ridge | $1.179 \pm 0.038$ | NN | $1.107 \pm 0.028$ |
| DS+MAML | $0.854 \pm 0.011$ | Lasso | $1.281 \pm 0.024$ | Mean | $1.347 \pm 0.025$ |
| NP | $0.845 \pm 0.012$ | BR | $1.544 \pm 0.134$ | | |

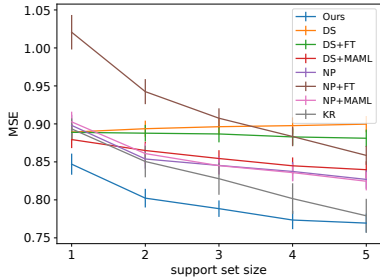

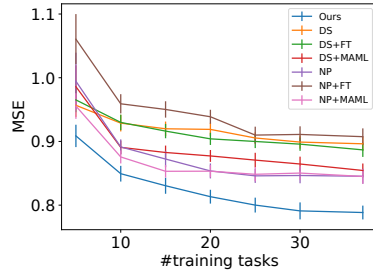

Figure 5: Averaged mean squared errors with different numbers of instances in a support set at a test phase with the OpenML datasets: The bar shows the standard error.

Figure 6: Averaged mean squared errors with different numbers of training tasks with the OpenML datasets: The bar shows the standard error.

Table 2: Training computational time in hours.

| Ours | DS | DS+FT | DS+MAML | NP | NP+FT | NP+MAML |
|---|---|---|---|---|---|---|
| 7.5 | 3.5 | 10.0 | 34.2 | 7.2 | 22.3 | 101.0 |

**Results** Table 1 shows the mean squared error averaged over 30 experiments with different training, validation, and target splits. The proposed method achieved the lowest error. DS+MAML outperformed DS and DS+FT, but it was worse than the proposed method. One of the reasons is that finding good initial parameters that quickly adapt to various tasks with heterogeneous attribute spaces was difficult with MAML. Since NP obtained representation for each task using a permutation invariant neural network, it could not explicitly model the characteristics for each attribute in a task. On the other hand, since the proposed method obtained representations for each attribute by effectively combining multiple permutation invariant neural networks, the proposed method achieved better performance. The mean squared error for each target task and the computational time is shown in the supplemental material. Table 2 shows the computational time for training 37 tasks on computers with 2.60GHz CPUs. The training time of the proposed method was shorter than MAML since the proposed method does not require iterative gradient descent steps for adapting to a support set. The computational time of the proposed method for testing 17 target tasks was 0.26 seconds. In the test phase, the proposed method efficiently predicted responses without optimization by feeding the support and query sets into the trained neural networks.

Figure 5 shows the averaged mean squared errors when we changed the number of instances in a support set at a test phase. In a training phase, models are trained with support set size $N_S = 3$. We omit the Ridge, Lasso, BR, GP, NN and Mean because of their high errors as shown in Table 1. With all methods except for DS, which did not use support sets, the error decreased as the support set size increased. The proposed method achieved the best in all cases.

Figure 6 shows the averaged mean squared errors with different numbers of training tasks. The error by the proposed method was the lowest with all numbers of training tasks. As the number of training tasks increased, the error decreased. By using more training tasks, tasks that resemble the target tasks are more likely to be included in the training. The proposed method learned a wide variety of patterns in the training tasks with different attribute spaces, and adequately used them for improving the performance on the target tasks.

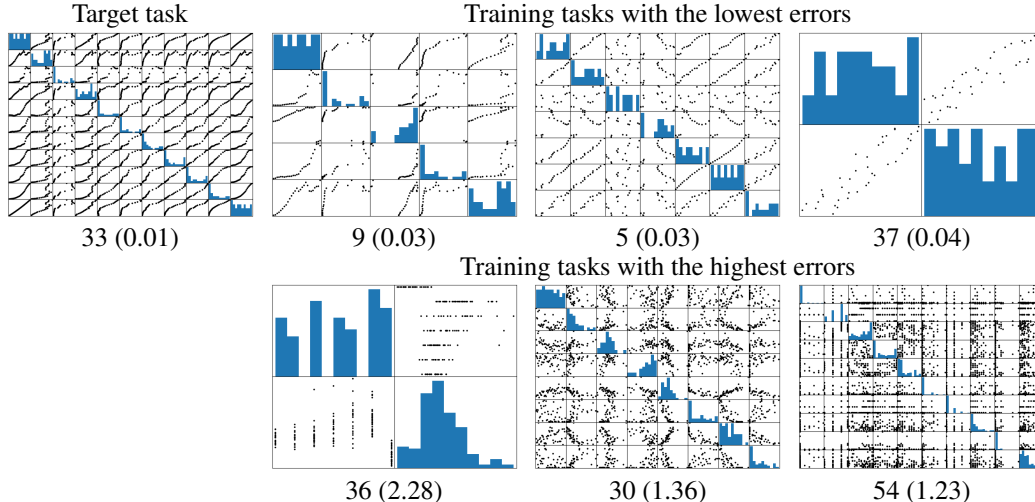

| Target task | Training tasks with the lowest errors | | |
|---|---|---|---|
| 33 (0.01) | 9 (0.03) | 5 (0.03) | 37 (0.04) |

Training tasks with the highest errors

| 36 (2.28) | 30 (1.36) | 54 (1.23) |
|---|---|---|

Figure 8: Scatter matrix plot for each task in the OpenML datasets: The horizontal and vertical axes are the attribute indexes. The left plot is the target task, and the three right plots at the top are the training tasks that achieved the lowest errors on the target task, and the three plots at the bottom are the training tasks that achieved the highest errors on the target task. The values below each plot are the task index and mean squared error of the target task when the task was used for training.

For analyzing the effectiveness of each task to improve the performance when used for training, we experimented with a single training task and a single target task for all task pairs. Figure 7 shows the result. Training tasks that improved the performance were different across the target tasks, and some training tasks deteriorated the performance depending on the target tasks. Figure 8 shows a scatter matrix plot of a target task and the training tasks with the lowest and highest errors on the target task. The training tasks with the lowest errors and the target task exhibit similar patterns. On the other hand, the training tasks with the highest errors exhibit different patterns from the target task. Since the proposed method can learn various patterns in different attribute spaces, the proposed method can exploit the knowledge learned from related tasks without being adversely influenced by unrelated tasks, which resulted in the lowest error when learned from multiple tasks as shown in Table 1.

For ablation study, we evaluated the proposed method when we changed the number of iterations of Eqs.(2,3) for obtaining the latent attribute vectors and latent response vectors in the inference network as described the last sentence in Section 3.2.1. Let $L$ be the number of iterations. When $L = 0$, the initial representations were used, i.e., $\mathbf{v}_i = \bar{\mathbf{v}}_i$ and $\mathbf{c}_i = \bar{\mathbf{c}}_i$ in Eq. (1) were used. The error with $L = 0$ in Table 3 was higher than the proposed method with $L = 1$ because the relationships across different attributes and responses were not encoded in the latent attribute and latent response vectors when $L = 0$. There was a small improvement from $L = 1$ to $L = 2$, but no further improvement after $L > 2$. We also evaluated the proposed method without sampling of attributes and responses for training, which was described in the second paragraph of Section 4.2. The error increased when the sampling was omitted as shown in Table 3 (w/o sampling). This result indicates that generating a wider variety of tasks by sampling attributes and responses is important to improve the performance on various target tasks.

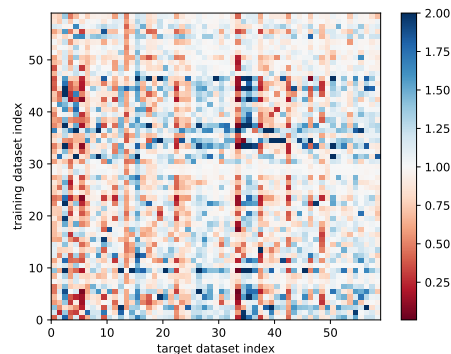

Figure 7: Averaged mean squared errors by the proposed method for each pair of training and target tasks with the OpenML datasets: The vertical axis is the training task index, and the horizontal axis is the target task index.

Table 3: Ablation study. The averaged mean squared errors on the target tasks with the OpenML datasets by the proposed method with the different number of iterations $L$ on Eqs. (2,3), and the proposed method without attribute and response sampling for training (w/o sampling).

| $L = 0$ | $L = 1$ | $L = 2$ | $L = 3$ | $L = 4$ | $L = 5$ | w/o sampling |
|---------|---------|---------|---------|---------|---------|--------------|
| 0.834 | 0.788 | 0.777 | 0.778 | 0.778 | 0.781 | 0.848 |

Table 4: Averaged accuracies and standard errors on the target tasks with the OpenML classification datasets. Values in bold typeface are not statistically different at 5% level from the best performing method in each row by a paired t-test.

| Method | Accuracy | Method | Accuracy | Method | Accuracy |
|--------|----------|--------|----------|--------|----------|
| Ours | $\mathbf{0.646 \pm 0.006}$ | NP+FT | $0.599 \pm 0.006$ | NN | $0.626 \pm 0.006$ |
| DS | $0.621 \pm 0.006$ | NP+MAML | $0.622 \pm 0.007$ | AB | $0.584 \pm 0.006$ |
| DS+FT | $0.616 \pm 0.006$ | KNN | $0.630 \pm 0.005$ | NB | $0.630 \pm 0.005$ |
| DS+MAML | $0.612 \pm 0.006$ | DT | $0.592 \pm 0.006$ | MF | $0.555 \pm 0.005$ |
| NP | $0.608 \pm 0.006$ | RF | $0.596 \pm 0.006$ | | |

**Classification**    We evaluated our proposed classification method with OpenML datasets, where the response was binarized depending on whether the value is above the mean or not for each task. We compared the proposed method with DS, DS+FT, DS+MAML, NP, NP+FT, NP+MAML, $k$-nearest neighbor method (KNN), decision tree (DT), random forest (RF), neural network (NN), Adaboost (AB), naive Bayes (NB), and most frequent class (MF), where KNN, DT, RF, NN, AB, NB, and MF were trained with the support set of the target task. Table 4 shows the accuracy on the target tasks averaged over 30 experiments. The proposed method achieved the highest accuracy.

## 5   Conclusion

We proposed a neural network-based meta-learning method that learns from multiple tasks with different attribute spaces, and predicts a response given a few instances in unseen tasks. Although we believe that our work is an important step for learning from a wide variety of tasks, we must extend our approach in several directions. First, we plan to improve the efficiency of the training procedure. Second, we will investigate different types of neural networks with variable length inputs for inferring latent attribute and response vectors, such as attentions [30, 24, 11, 16]. Third, we want to extend the proposed method to use prior knowledge about attributes, such as correspondence information across tasks and descriptions on attributes.

## Broader impact

The proposed method improves regression/classification performance even when a small number of labeled data are given by training from various tasks whose attribute spaces are different from the target tasks. The proposed method can be used for applications where machine learning has not been used due to the scarcity of labeled data. The proposed method could reduce the cost of manual labeling for increasing labeled data. Since the proposed method can use various datasets with heterogeneous attribute spaces for training, there is a potential risk that users might include biased datasets without careful thought in training datasets, which might result in biased predictions. We encourage research to automatically detect biased datasets. Although the proposed method improves performance, the prediction might be not always correct. We encourage research to estimate the uncertainty of the prediction by meta-learning from tasks with heterogeneous attribute spaces.

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
