[Supplementary Material]

# Supplemental Material: Meta-learning from Tasks with Heterogeneous Attribute Spaces

**Tomoharu Iwata**
NTT Communication Science Laboratories
tomoharu.iwata.gy@hco.ntt.co.jp

**Atsutoshi Kumagai**
NTT Software Innovation Center
atsutoshi.kumagai.ht@hco.ntt.co.jp

## 1 Experiments

### 1.1 OpenML data

**Data**   Table 1 shows the statistics of the OpenML datasets.

**Comparing methods**   DS predicted responses by $\hat{y} = g_{\text{DS}}(\frac{1}{I} \sum_{i=1}^{I} f_{\text{DS}}(x_i))$, where $g_{\text{DS}}$ and $f_{\text{DS}}$ were three-layered feed-forward neural networks with 32 hidden units, and the number of output units of $f_{\text{DS}}$ was 32. DS, which can handle different numbers of attributes, was trained to minimize the loss on the training datasets.

With NP, we used deep sets for handling tasks with heterogeneous attribute spaces. In particular, the attributes were encoded in a 32-dimensional vector by a deep set, and the responses were encoded in a 32-dimensional vector by a neural network. Then the two encoded vectors were concatenated and input to a neural network. By averaging the outputs over the instances in the support set, we obtained a task representation. The task representation and each attribute value was concatenated, and the response was predicted by a deep set by taking the set of concatenated vectors as input. In the NP, we used three-layered feed-forward neural networks with 32 hidden units for all the neural networks, and average pooling for all the deep sets.

DS+FT (NP+FT) was the DS (NP) fine-tuned with each target dataset. With DS+MAML (NP+MAML), DS (NP) was trained to minimize the loss on the query sets when fine-tuned with the support sets with model-agnostic meta-learning [1], where support and query sets were sampled in the same way with the proposed method in Algorithm 1. The number of fine-tuning epochs was five.

Ridge, Lasso, BR, KR, GP, NN, and Mean were trained with the target datasets since they cannot handle tasks with heterogeneous attribute spaces. We used the implementation of scikit-learn [2] for them, where the default parameter settings of the scikit-learn were used since the number of support instances was three, which is too small to conduct a cross-validation.

For classification experiments, we used the cross-entropy loss with DS, DS+FT, DS+MAML, NP, NP+FT, NP+MAML, and the proposed method. We used the implementation of scikit-learn [2] with KNN, DT, RF, NN, AB, NB, and MF, where the default parameter settings of the scikit-learn were used.

**Results**   Table 2 shows the mean squared error for each target task.

## References

[1] E. Grefenstette, B. Amos, D. Yarats, P. M. Htut, A. Molchanov, F. Meier, D. Kiela, K. Cho, and S. Chintala. Generalized inner loop meta-learning. *arXiv preprint arXiv:1910.01727*, 2019.

Table 1: Statistics of the OpenML datasets used in our experiments.

| Index | Name | #instances | #attributes |
|---|---|---|---|
| 0 | vineyard | 52 | 3 |
| 1 | bolts | 40 | 7 |
| 2 | sleep | 62 | 8 |
| 3 | autoPrice | 159 | 16 |
| 4 | detroit | 13 | 14 |
| 5 | longley | 16 | 7 |
| 6 | diabetes_numeric | 43 | 3 |
| 7 | baskball | 96 | 5 |
| 8 | pyrim | 74 | 28 |
| 9 | gascons | 27 | 5 |
| 10 | pwLinear | 200 | 11 |
| 11 | machine_cpu | 209 | 7 |
| 12 | qsbr_y2 | 25 | 10 |
| 13 | qsfsr1 | 20 | 10 |
| 14 | qsfsr2 | 19 | 10 |
| 15 | qsbralks | 13 | 22 |
| 16 | qsartox | 16 | 24 |
| 17 | qsabr2 | 15 | 10 |
| 18 | hip | 54 | 8 |
| 19 | analcatdata_uktrainacc | 31 | 16 |
| 20 | mu284 | 284 | 10 |
| 21 | pollution | 60 | 16 |
| 22 | transplant | 131 | 3 |
| 23 | bodyfat | 252 | 15 |
| 24 | fri_c0_250_5 | 250 | 6 |
| 25 | fri_c3_100_10 | 100 | 11 |
| 26 | fri_c2_100_5 | 100 | 6 |
| 27 | fri_c3_250_10 | 250 | 11 |
| 28 | fri_c2_250_25 | 250 | 26 |
| 29 | fri_c4_100_25 | 100 | 26 |
| 30 | sleuth_ex1714 | 47 | 8 |
| 31 | rabe_265 | 51 | 7 |
| 32 | rabe_266 | 120 | 3 |
| 33 | chscase_demand | 27 | 11 |
| 34 | visualizing_slope | 44 | 4 |
| 35 | visualizing_environmental | 111 | 4 |
| 36 | chscase_funds | 185 | 2 |
| 37 | rabe_166 | 40 | 2 |
| 38 | sleuth_ex1605 | 62 | 6 |
| 39 | chscase_vine1 | 52 | 10 |
| 40 | rabe_131 | 50 | 6 |
| 41 | diggle_table_a1 | 48 | 5 |
| 42 | chatfield_4 | 235 | 13 |
| 43 | hutsof99_child_witness | 42 | 16 |
| 44 | rabe_176 | 70 | 4 |
| 45 | visualizing_hamster | 73 | 6 |
| 46 | rabe_148 | 66 | 6 |
| 47 | visualizing_ethanol | 88 | 3 |
| 48 | chscase_geyser1 | 222 | 3 |
| 49 | humans_numeric | 75 | 15 |
| 50 | USCrime | 47 | 14 |
| 51 | ICU | 200 | 20 |
| 52 | EgyptianSkulls | 150 | 5 |
| 53 | heart | 270 | 14 |
| 54 | treepipit | 86 | 10 |
| 55 | edm | 154 | 18 |
| 56 | slump | 103 | 10 |
| 57 | branin | 225 | 3 |
| 58 | echocardiogram-uci | 132 | 8 |

[2] F. Pedregosa, G. Varoquaux, A. Gramfort, V. Michel, B. Thirion, O. Grisel, M. Blondel, P. Prettenhofer, R. Weiss, V. Dubourg, et al. Scikit-learn: Machine learning in Python. *Journal of Machine Learning Research*, 12:2825–2830, 2011.

Table 2: Averaged mean squared errors and standard errors on each target task with the OpenML datasets: Values in bold typeface are not statistically different at 5% level from the best performing method in each row according to a paired t-test. The bottom row is the number of tasks where the method achieved the performance that was not statistically different from the best performing method.

| Index | Ours | DS | DS+FT | DS+MAML | NP | NP+FT | NP+MAML | Ridge | Lasso | BR | KR | GP | NN | Mean |
|---|---|---|---|---|---|---|---|---|---|---|---|---|---|---|
| 0 | 0.827 | 0.494 | 0.502 | 0.516 | 0.472 | 0.548 | **0.463** | 0.727 | 1.188 | 0.623 | 0.574 | 1.175 | 0.917 | 1.270 |
| 1 | 0.786 | 0.884 | 0.903 | 0.895 | 0.886 | 0.876 | 0.868 | 1.362 | 1.490 | 1.386 | **0.512** | 0.903 | 0.869 | 1.501 |
| 2 | **0.705** | 1.737 | 1.614 | 1.177 | 1.051 | 0.776 | 0.920 | 1.188 | 1.245 | 1.791 | 0.851 | 0.997 | 1.320 | 1.217 |
| 3 | 0.710 | 0.763 | 0.731 | 0.556 | 0.485 | 0.517 | 0.552 | 0.821 | 1.204 | 0.769 | **0.414** | 0.962 | 0.855 | 1.272 |
| 4 | 0.699 | 0.787 | 0.794 | 0.716 | 0.666 | 0.695 | 0.647 | 1.095 | 1.407 | 1.414 | **0.453** | 0.931 | 0.683 | 1.521 |
| 5 | 0.758 | 0.362 | 0.306 | 0.182 | 0.140 | 0.198 | 0.132 | 0.267 | 0.925 | 0.258 | **0.084** | 0.530 | 0.313 | 1.138 |
| 6 | 0.689 | 0.667 | 0.675 | 0.697 | **0.662** | 0.771 | 0.675 | 1.410 | 1.488 | 2.445 | 0.906 | 1.102 | 1.305 | 1.505 |
| 7 | **0.817** | 0.912 | 0.929 | 0.962 | 0.980 | 1.117 | 0.955 | 1.346 | 1.557 | 1.551 | 1.062 | 1.024 | 1.380 | 1.560 |
| 8 | 0.930 | 0.919 | 0.917 | 0.891 | **0.854** | 0.894 | 0.901 | 1.180 | 1.232 | 1.182 | 0.952 | 1.016 | 1.250 | 1.232 |
| 9 | 0.707 | 0.564 | 0.471 | 0.407 | 0.405 | 0.414 | 0.459 | 0.477 | 1.209 | 4.763 | 0.355 | **0.311** | 0.349 | 1.240 |
| 10 | **0.834** | 0.949 | 0.979 | 0.980 | 0.964 | 1.055 | 0.986 | 1.355 | 1.477 | 1.353 | 0.842 | 0.992 | 1.191 | 1.477 |
| 11 | 0.822 | 0.440 | 0.378 | 0.317 | 0.279 | 0.302 | **0.272** | 0.577 | 0.971 | 0.520 | 0.375 | 0.897 | 0.553 | 1.551 |
| 12 | **0.803** | 1.175 | 1.178 | 1.105 | 1.031 | 1.101 | 1.044 | 1.304 | 1.373 | 1.607 | 1.169 | 0.966 | 1.311 | 1.434 |
| 13 | **0.810** | 1.162 | 1.173 | 1.142 | 1.199 | 1.234 | 1.173 | 1.316 | 1.378 | 1.333 | 1.148 | 1.057 | 1.154 | 1.367 |
| 14 | 0.825 | 1.232 | 1.233 | 1.211 | 1.268 | 1.301 | 1.276 | 1.060 | 1.355 | 1.029 | **0.724** | 1.056 | 0.967 | 1.355 |
| 15 | **0.728** | 1.100 | 1.115 | 1.154 | 1.095 | 1.136 | 1.096 | 1.838 | 1.120 | 2.129 | 1.210 | 1.073 | 1.798 | 1.142 |
| 16 | 0.931 | 0.930 | 0.950 | 0.892 | 0.861 | 0.948 | 0.868 | 1.115 | 1.457 | 1.101 | **0.725** | 1.067 | 1.207 | 1.482 |
| 17 | **0.567** | 1.106 | 1.142 | 1.177 | 1.145 | 1.323 | 1.152 | 1.007 | 1.257 | 1.023 | 0.644 | 1.018 | 0.973 | 1.300 |
| 18 | 0.831 | 0.904 | 0.913 | 0.907 | 0.940 | 1.038 | 0.938 | 0.963 | 1.095 | 0.858 | **0.727** | 0.869 | 1.093 | 1.135 |
| 19 | 0.723 | 0.548 | 0.559 | 0.585 | **0.547** | 0.629 | 0.580 | 0.968 | 1.014 | 0.959 | 0.606 | 0.903 | 1.169 | 1.621 |
| 20 | 0.780 | 1.210 | 1.170 | 1.221 | 1.194 | 1.327 | 1.224 | 0.978 | 1.190 | 1.130 | **0.731** | 0.833 | 1.066 | 1.242 |
| 21 | **0.688** | 0.969 | 0.984 | 0.977 | 0.965 | 1.061 | 0.976 | 1.116 | 1.340 | 1.192 | 0.812 | 0.978 | 0.934 | 1.355 |
| 22 | 0.785 | 0.457 | 0.371 | 0.264 | 0.223 | 0.362 | **0.212** | 0.516 | 1.155 | 0.712 | 0.306 | 0.644 | 0.564 | 1.234 |
| 23 | 0.744 | 0.746 | 0.758 | 0.762 | 0.738 | 0.847 | 0.801 | 1.080 | 1.092 | 1.135 | **0.463** | 0.974 | 0.695 | 1.209 |
| 24 | 0.795 | 0.709 | 0.678 | 0.660 | 0.686 | **0.620** | 0.630 | 1.005 | 1.194 | 1.029 | 0.790 | 0.908 | 0.889 | 1.224 |
| 25 | **0.834** | 0.993 | 1.000 | 1.006 | 1.012 | 1.067 | 1.018 | 1.412 | 1.187 | 1.576 | 1.319 | 1.006 | 1.425 | 1.185 |
| 26 | **0.752** | 1.278 | 1.226 | 1.045 | 1.151 | 1.182 | 1.057 | 1.451 | 1.078 | 1.732 | 1.293 | 0.947 | 1.411 | 1.051 |
| 27 | **0.676** | 0.986 | 0.980 | 0.990 | 1.019 | 1.148 | 1.020 | 1.434 | 1.097 | 1.560 | 1.264 | 0.994 | 1.324 | 1.098 |
| 28 | **0.797** | 1.061 | 1.068 | 1.044 | 1.045 | 1.144 | 1.030 | 1.184 | 1.199 | 1.190 | 1.081 | 1.001 | 1.323 | 1.183 |
| 29 | **0.660** | 1.014 | 1.025 | 1.020 | 1.007 | 1.030 | 1.011 | 1.323 | 1.337 | 1.332 | 0.943 | 0.997 | 1.169 | 1.292 |
| 30 | 0.706 | 1.000 | 1.015 | 1.027 | 1.038 | 1.112 | 1.028 | 0.989 | 1.275 | 1.251 | **0.523** | 0.881 | 0.664 | 1.284 |
| 31 | **0.834** | 0.929 | 0.930 | 0.954 | 0.956 | 1.069 | 0.955 | 1.921 | 1.488 | 2.271 | 1.509 | 1.125 | 1.747 | 1.351 |
| 32 | 1.014 | 0.810 | 0.821 | 0.874 | 0.833 | 0.996 | 0.853 | 0.481 | 1.283 | **0.117** | 0.185 | 0.482 | 0.330 | 1.432 |
| 33 | 0.718 | 0.310 | 0.215 | 0.090 | 0.050 | 0.048 | 0.067 | 0.200 | 1.384 | 0.086 | **0.021** | 0.575 | 0.218 | 1.506 |
| 34 | 0.617 | 1.619 | 1.560 | 1.371 | 1.398 | 1.211 | 1.367 | 0.759 | 1.119 | 0.888 | **0.521** | 0.664 | 0.749 | 1.142 |
| 35 | **0.781** | 1.740 | 1.688 | 1.401 | 1.316 | 1.194 | 1.335 | 1.086 | 1.272 | 1.864 | 1.023 | 1.097 | 1.331 | 1.321 |
| 36 | **0.878** | 1.282 | 1.184 | 1.121 | 1.180 | 1.215 | 1.180 | 2.062 | 1.462 | 4.445 | 1.257 | 1.846 | 2.200 | 1.462 |
| 37 | 0.969 | 0.406 | 0.368 | **0.252** | 0.314 | 0.360 | 0.280 | 0.773 | 1.209 | 1.273 | 0.305 | 8.921 | 0.521 | 1.283 |
| 38 | 0.747 | 0.708 | 0.735 | 0.725 | **0.702** | 0.777 | 0.718 | 1.109 | 1.249 | 1.173 | 0.814 | 1.012 | 1.075 | 1.252 |
| 39 | 0.834 | **0.758** | 0.778 | 0.831 | 0.888 | 0.969 | 0.881 | 1.102 | 1.170 | 1.263 | 0.843 | 0.989 | 1.137 | 1.194 |
| 40 | 0.853 | 0.752 | 0.757 | 0.839 | 0.862 | 1.048 | 0.869 | 0.724 | 1.153 | 0.832 | **0.423** | 0.732 | 0.692 | 1.368 |
| 41 | **0.721** | 0.839 | 0.857 | 0.875 | 0.855 | 0.958 | 0.862 | 1.577 | 1.474 | 2.118 | 1.221 | 1.009 | 1.420 | 1.488 |
| 42 | 0.950 | 0.390 | 0.368 | **0.297** | 0.299 | 0.333 | 0.303 | 0.511 | 1.125 | 0.483 | 0.328 | 0.851 | 0.620 | 1.156 |
| 43 | **0.782** | 0.894 | 0.901 | 0.946 | 0.912 | 1.050 | 0.934 | 1.539 | 1.117 | 1.643 | 1.026 | 0.993 | 1.284 | 1.232 |
| 44 | **0.642** | 0.823 | 0.826 | 0.934 | 0.978 | 1.289 | 0.964 | 2.305 | 1.318 | 2.123 | 1.341 | 1.559 | 2.151 | 1.242 |
| 45 | **0.845** | 1.113 | 1.149 | 1.102 | 1.033 | 1.082 | 1.023 | 2.143 | 1.675 | 3.024 | 1.422 | 1.161 | 2.003 | 1.675 |
| 46 | 0.746 | 0.540 | 0.528 | **0.416** | 0.491 | 0.694 | 0.588 | 1.257 | 1.347 | 1.431 | 0.609 | 0.947 | 1.024 | 1.446 |
| 47 | **0.774** | 1.195 | 1.211 | 1.078 | 1.130 | 1.270 | 1.101 | 2.037 | 1.467 | 2.648 | 1.519 | 1.950 | 2.695 | 1.461 |
| 48 | 0.828 | 0.704 | 0.702 | 0.726 | 0.700 | 0.840 | 0.705 | 0.776 | 1.177 | 1.227 | **0.380** | 0.760 | 0.753 | 1.238 |
| 49 | **0.735** | 1.115 | 1.135 | 1.178 | 1.183 | 1.241 | 1.184 | 1.694 | 1.191 | 2.103 | 1.722 | 1.098 | 1.559 | 1.183 |
| 50 | 0.819 | 1.141 | 1.135 | 1.060 | 1.081 | 1.123 | 1.054 | 0.848 | 1.154 | 0.883 | **0.509** | 0.927 | 0.649 | 1.412 |
| 51 | **0.861** | 0.976 | 0.999 | 0.988 | 0.961 | 1.031 | 0.968 | 1.071 | 1.104 | 1.071 | 1.006 | 1.008 | 1.257 | 1.118 |
| 52 | **0.828** | 1.111 | 1.113 | 1.068 | 1.050 | 1.109 | 1.051 | 1.756 | 1.389 | 2.434 | 1.191 | 1.030 | 1.473 | 1.378 |
| 53 | **0.660** | 0.816 | 0.810 | 0.741 | 0.754 | 0.772 | 0.758 | 1.528 | 1.664 | 1.552 | 0.748 | 0.990 | 1.061 | 1.664 |
| 54 | **0.934** | 1.048 | 1.056 | 1.050 | 1.044 | 1.065 | 1.033 | 2.262 | 2.039 | 2.832 | 1.450 | 0.985 | 1.563 | 2.039 |
| 55 | **0.857** | 0.984 | 0.982 | 1.011 | 1.000 | 1.039 | 1.004 | 1.755 | 1.116 | 2.047 | 1.105 | 1.023 | 1.311 | 1.141 |
| 56 | **0.767** | 1.061 | 1.061 | 1.041 | 1.039 | 1.066 | 1.046 | 1.441 | 1.398 | 1.486 | 1.010 | 0.973 | 1.207 | 1.397 |
| 57 | **0.775** | 1.011 | 0.959 | 1.050 | 1.075 | 1.119 | 1.087 | 1.353 | 1.253 | 5.520 | 1.173 | 0.796 | 1.161 | 1.253 |
| 58 | 0.924 | **0.795** | 0.813 | 0.814 | 0.811 | 0.902 | 0.804 | 1.331 | 1.384 | 1.393 | 1.098 | 0.980 | 1.271 | 1.495 |
| #best | **29** | 2 | 0 | 3 | 4 | 1 | 3 | 0 | 0 | 1 | 15 | 1 | 0 | 0 |