[Reviews · NeurIPS 2020]

Review 1

Summary and Contributions: The paper provides a neural network architecture for few-shot learning for heterogeneous attribute spaces. Via experiments on a synthetic dataset and OpenML datasets, they demonstrate superior performance as compared to baselines in regression and classification tasks which involves heterogeneous feature spaces.

Strengths: (i) First of its kind of work which explicitly addresses the heterogeneous feature space few-shot learning problem. (ii) The model proposed draws inspiration from recent works which are based on developing models for permutation-invariant set representations. These models have potential for the few-shot and meta-learning setting, though I found this connection to be not stated properly in the paper.

Weaknesses: (i) Missing References and Comparisons: Comparison (and citations) with CNAP [1] and self-attention based approaches [2] should be included. Self-attention is itself permutation-invariant (unless you use positional encoding). In a way, self-attention "generalises" the summation operation as it performs a weighted summation of different attention vectors. By setting all keys and queries to 1.0, you effectively end up with the Deep Sets architecture. I also feel a comparison with Prototypical Nets [3] for the few-shot classification setting is needed seeing its close resemblance of the way latent attribute vectors are calculated. (ii) Bechmarks: Results on more realistic data benchmarks like Meta-Dataset [4] or the Hetro-lingual text classification dataset used by Reference [19] of the paper ( Section 4.3) would be interesting. I feel that showing results on synthetic data and OpenML data ( relatively easy) is not that interesting. (iii) I feel details like what do the baselines like NP + FT, DS + MAML mean are important and should be included in the main paper rather than the supplementary material. It was really difficult for me before to understand what they meant and how they were adapted to the heterogeneous attribute space setting. (iv) It seems the method is computationally expensive seeing that it has to calculate the latent attribute vector in multiple loops. Some mention of computational complexity of the method as compared to baselines should be included for reference purposes. [1] Requeima, James, Jonathan Gordon, John Bronskill, Sebastian Nowozin, and Richard E. Turner. "Fast and flexible multi-task classification using conditional neural adaptive processes." In Advances in Neural Information Processing Systems, pp. 7959-7970. 2019. [2] Ashish Vaswani, Noam Shazeer, Niki Parmar, Jakob Uszkoreit, Llion Jones, Aidan N Gomez, Lukasz Kaiser, and Illia Polosukhin. Attention is all you need. arXiv preprint arXiv:1706.03762, 2017. [3] Snell, Jake, Kevin Swersky, and Richard Zemel. "Prototypical networks for few-shot learning." In Advances in neural information processing systems, pp. 4077-4087. 2017. [4] https://github.com/google-research/meta-dataset *******************After Rebuttal ******************************* The rebuttal carifies some of my concerns and I am satisfied by their answers to some of my questions. I have decided to increase my rating to 6.

Correctness: Yes.

Clarity: The writing is fine but certain details experimental details can be included. See Comments later.

Relation to Prior Work: Yes. I do feel certain references and connections and comparisons with them are missing. See Weaknesses above.

Reproducibility: Yes

Additional Feedback: Questions: (i) Line 263: There was a small improvement from L = 1 to L = 2, but no further improvement after L > 2. Why? (ii) Line 219: One of the reasons is that finding parameters that quickly adapt to various tasks with heterogeneous attribute spaces was difficult with MAML. Why? Comments: (i) Figure 7 conveys no extra information and can be removed. (ii) The baselines and why they were chosen should be described more in the main paper. (iii) The usefulness of permutation-invariant approaches like deep sets and CNPs for meta-learning can be stated more explicitly in Related Works section.


Review 2

Summary and Contributions: This paper proposes a heterogeneous meta-learning method that trains a model on tasks with various attribute spaces, and tests on unseen few-shot tasks whose attribute spaces are different. Experiments are conducted with synthetic datasets and 59 datasets in OpenML.

Strengths: The problem and methodology are new. Meta-learning with heterogeneous attribute spaces hasn’t been studied before. It is good to extend meta-learning into this scenario.

Weaknesses: 1. The description didn’t separate the meta-learning parameters from the task specific parameters. At least the f functions would be task specific. How would one train the task specific parameters on the unseen test tasks? [-- after rebuttal: The meta-training just provides initial parameters. ] 2. It also lacks sufficient motivation and justification on why doing so. Do the inference network and prediction network together form a good prediction model? This needs to be justified on regular supervised learning setting with varying number of training instances. 3. As there are no previous methods on this setting, to ensure a fair comparison, the authors can compare to standard meta-learning methods on standard meta-learning datasets. Though there are no heterogeneous attributes, such experiments can clarify whether the proposed model sacrifices prediction performance to exchange for the complexity of handling heterogeneous attribute spaces.

Correctness: In 3.2, the latent attribute vectors seem to be the mean vectors of the support instances in transformed feature space, while the latent response vectors seem to be the mean vectors of the support responses in transformed feature space. It is not clear why should one compute the instance representation using Eq.(2). Note f_u function is applied on both [\bar{v},x] and [\bar{c},y], which are not even in the same feature space and does not even have the same feature dimension. This is problematic. --After reading the rebuttal, I agree the dimension for [\bar{v},x] and [\bar{c},y] can be set as the same.

Clarity: It is well written in general. Clarity can be improved.

Relation to Prior Work: clear

Reproducibility: No

Additional Feedback:


Review 3

Summary and Contributions: This paper proposes a novel approach to meta-learning with heterogeneous attributes spaces across tasks. It introduces a network architecture based on deep set operators that can handle various number of attributes, responses, and instances in a task. Comprehensive comparison with baseline methods are done using OpenML datasets with superior performance in both regression and classification.

Strengths: This is the first meta-learning approach to tasks with heterogeneous attribute spaces. The network architecture is well designed and ablated. It is particularly nice to design various meta-learning baselines based on deep set and neural processes and conduct thorough comparison with them together with standard ML methods. The experiments on the OpenML datasets are convincing and the detailed analysis is helpful for understanding the proposed method's behaviour. Using attention method to handle variable length of instances and attributes is a straightforward and promising direction. I'm glad the authors are planning to explore that in future work.

Weaknesses: The artificial construction of the regression and classification tasks from OpenML may seem a little bit different from real application scenarios. It would be helpful to consider a real application, where learning from other tasks of different attribute spaces is useful to solve the real problem. Scalability on the attribute and response dimensions is still to address. Default scikit-learn hyper-parameters are used for standard ML methods. Although training tasks can not be used for these methods for meta-learning, they can be used at least to search for a better set of hyper-parameters. It seems the KR and NN may have the potential of matching or exceeding the proposed method in regression and classification respectively, especially when the size of support set increases. The last point is probably not a weakness of the paper, but I was wondering if prior knowledge existed about which subset of attributes were shared among pairs of tasks, could one have a better meta-learning algorithm to make use of that? This would be useful in address multi-modality learning problems.

Correctness: The proposed method is reasonable, and the empirical evaluation

Clarity: This paper is well written. The network architecture and training procedure are clearly explained.

Relation to Prior Work: I'm not particularly familiar with the literature of handling heterogeneous attributes.

Reproducibility: Yes

Additional Feedback:


Review 4

Summary and Contributions: The paper introduces the first approach to meta-learning from heterogeneous attribute spaces.

Strengths: The proposed approach appears to be novel and is quite elegant in its formulation.

Weaknesses: The paper suffers from two main weaknesses: 1) it has no real-world motivating problem, which makes the reading extremely painful, as the reader keeps trying to imagine an intuitive scenario in which to apply this approach 2) the paper introduces a complex approach that, on the categorization problem, barely outperforms kNN. If this difference in performance was on a real world problem and it translated into saved lives or reductions in costs, the choice of may be obvious. For an artificially-created problem, it is difficult to justify NOT using kNN instead. The regression results are better, but the baselines were not designed to solve this problem.

Correctness: The claims appear to be correct.

Clarity: The paper is reasonably well written, but, especially in the first two pages, it suffers from a lack of an intuitive, real-world, motivating example. Even if there is no corresponding dataset in the evaluation, the paper should use such an example (and maybe even a running example on how the proposed approach works). For example, even the Abstract is quite cryptic because there is no intuitive, well-know problem to map it to. The authors could follow the example of [32], where the application domains are introduced from the abstract.

Relation to Prior Work: yes

Reproducibility: Yes

Additional Feedback:

[Author Response · NeurIPS 2020]

We would like to thank the reviewers for their feedback and comments, which we shall address below.

**To Reviewer1**

**> Missing References and Comparisons:** As you commented, and as written at the last sentence in the conclusion
section, attention-based approaches can be used in our framework. Our contribution is not to develop permutation
invariant networks, but to develop a few-shot learning method for heterogeneous attribute spaces using permutation
invariant networks. Since Prototypical nets cannot handle heterogeneous attribute spaces, we did not compare with
them. The computational time (hours) were: Ours: 7.5, DS:3.5, DS+FT:10.0, DS+MAML:34.2, NP:7.2, NP+FT:22.3,
NP+MAML:101.0. We will include the missing references and computational complexity of baselines.

**> Results on more realistic data benchmarks:** Meta-Dataset is image data, and Hetro-lingual text classification
dataset is text data. Their attribute sizes might be different, but the modality is shared. On the other hand, OpenML data
contains datasets with different modality. Since our aim is to develop a model that can be learned from any datasets, we
believe that OpenML is more suitable.

**> quickly adapt to various tasks with heterogeneous attribute spaces was difficult with MAML. Why?:** MAML
learns good initial parameters that achieve good performance when finetuned. Good initial parameters would be
different across various tasks with different attributes.

**To Reviewer 2**

**> How would one train the task specific parameters on the unseen test tasks?** By taking the support set as input,
we can obtain the task specific parameters on the unseen test task using the neural networks that are shared across all
tasks. Some meta-learning methods (e.g., matching networks and conditional neural processes) also use shared neural
networks to obtain the task specific parameters.

**> Do the inference network and prediction network together form a good prediction model?:** The inference
network infers the task specific parameters given the support set, which can be seen as training on regular supervised
learning, where the training procedure is approximated by the neural networks. The prediction network predicts a
response of an instance using the task specific parameters.

**> compare to standard meta-learning methods on standard meta-learning datasets:** We compared with standard
meta-learning methods, MAML and NP, with heterogeneous datasets as written in our experiments. The standard
meta-learning methods on standard meta-learning datasets are not fair since the standard meta-learning methods know
that their attribute spaces are the same, but the proposed method does not know.

**> $[\bar{\mathbf{v}}_i, x_{ni}]$ and $[\bar{\mathbf{c}}_j, y_{nj}]$, which are not even in the same feature space and does not even have the same feature**
**dimension:** $x_{ni}$ and $y_{ni}$ in Eq(2) are scalar values. $\bar{\mathbf{v}}_i$ and $\bar{\mathbf{c}}_j$ are the outputs of neural netowks $g$ in Eq(1), and their
dimensions are the same by using neural networks with the same output unit size. Therefore, $[\bar{\mathbf{v}}_i, x_{ni}]$ and $[\bar{\mathbf{c}}_j, y_{nj}]$
have the same dimension. We can use different functions $f_u$ for $[\bar{\mathbf{v}}_i, x_{ni}]$ and $[\bar{\mathbf{c}}_j, y_{nj}]$, but used the same function for
simplicity. We used Eq(2) to calculate the instance representation using all attributes and all responses.

**To Reviewer3**

**> The artificial construction of the regression and classification:** We admit that the classification task in our task is
a bit artificial. But, we included the classification experiments to demonstrate that the proposed method is applicable to
classification tasks. The regression experiments with OpenML demonstrates that our method can learn from various
datasets.

**> if prior knowledge existed about which subset of attributes were shared among pairs of tasks:** Yes. We think
the proposed method can be improved by sharing attribute representations for shared attributes.

**To Reviewer 4**

**> it has no real-world motivating problem:** We want to develop a model that can be learned from any datasets, and
that can be used for an unseen task. For example, consider anomaly detection for various machines in various factories.
Attributes (e.g., sensors) are different across machines. But, there are related machines. We want to detect anomalies
for a new machine in a new factory with only a few labeled data, by utilizing data of existing machines. We include
real-world motivating examples.

**> complex approach:** Although the difference of the performance between the proposed method and kNN was not
large in our classification experiment, the performance of the proposed method was statistically better than that of kNN.
We believe that our work is an important step for learning from a wide variety of datasets. Since there are no existing
methods for solving this problem, we used the baselines that were not designed to solve the problem.

[Meta-Review · NeurIPS 2020]

This paper proposed a method for shot learning in heterogeneous attribute spaces, and shows that it performs well in a set of evaluations from synthetic tasks and OpenML datasets. Although 75% of the reviewers were leaning towards reject initially, the rebuttal and reviewer 3 convinced the dissenting majority (in fact, all reviewers who participated in the discussion) to lean towards supporting acceptance. The paper has a few outstanding weaknesses which can in part be addressed by revising the writing, and in part in future work, but for now it sufficiently proves the concept in a new area to warrant publication.